# Sodium Butyrate Alleviates Mouse Colitis by Regulating Gut Microbiota Dysbiosis

**DOI:** 10.3390/ani10071154

**Published:** 2020-07-07

**Authors:** Xiujing Dou, Nan Gao, Di Yan, Anshan Shan

**Affiliations:** Institute of Animal Nutrition, Northeast Agricultural University, Harbin 150030, China; douxiujing@neau.edu.cn (X.D.); 15663686078@163.com (N.G.); yandi163com@163.com (D.Y.)

**Keywords:** sodium butyrate, colitis, microbiota

## Abstract

**Simple Summary:**

Inflammatory bowel disease (IBD) is extremely harmful to animal health and can affect animal growth and production. Dextran sulfate sodium (DSS) can cause IBD in animals, resulting in diarrhea and bloody stools. The present study aimed to determine whether oral sodium butyrate can relieve DSS-induced colitis in mice. By using histological evaluation (H&E) staining technology and 16S rRNA sequence analysis, we found that the severity of colitis in mice receiving oral sodium butyrate was reduced, and the composition of gut microbiota was changed. These results indicate that sodium butyrate can relieve DSS-induced colitis in mice by restoring the balance of gut microbiota dysbiosis.

**Abstract:**

Inflammatory bowel disease (IBD) develops as a result of complicated interactions between genetic susceptibility, excessive innate immunity, and environmental factors, which are mainly related to the gut microbiota. The present study aimed to elucidate the protective effects and underlying mechanisms of a short-chain fatty acid salt, sodium butyrate, on colonic inflammation induced by dextran sulfate sodium (DSS) in mice. Pretreatment with sodium butyrate attenuated colitis, as demonstrated by the decreased disease activity index (DAI), colon length shortening, spleen tumidness, and histopathology scores, while maintaining intestinal barrier integrity, as observed by H&E staining and electron microscopy. 16S rRNA sequence analysis revealed that sodium butyrate caused a remarkable alteration of the gut microbiota. *Bacteroides*, *Lachnospiraceae*, the *Lachnospiraceae NK4A136*
*group*, and *Ruminiclostridium 6* presented dramatic differences after sodium butyrate supplementation. This work verifies that sodium butyrate can improve mouse colitis via microbe–host interactions by regulating the microbial community. Taken together, the findings demonstrate that sodium butyrate shows great potential as a probiotic agent for ameliorating colitis.

## 1. Introduction

Inflammatory bowel disease (IBD) is an autoimmune disease characterized by a chronic inflammatory status resulting from an abnormal immune response. Accumulating data have shown that the pathological mechanisms of IBD are closely related to intestinal microflora imbalances, epithelial barrier dysfunction, and tissue damage [1,2]. The immune system and microbiota of the intestine maintain intestinal homeostasis to improve intestinal development [3]. Dysbiosis in the colon accelerates the progression of colitis, and the microbial community of IBD patients is different from that of healthy individuals [4,5]. The incidence of IBD has previously been related to dietary patterns and additives that modulate the gut microbiota [6,7,8].

Short-chain fatty acids (SCFAs), which are mainly produced by the cecal fermentation of nonstarch polysaccharides, play crucial roles in maintaining colon pH and osmolarity [9]. They are absorbed and efficiently utilized in the intestinal mucosa and exhibit many physiological functions in the host intestine, including providing energy, regulating gene expression, and serving as signaling molecules that promote gut health [10,11,12]. Butyrate, one of the main four-carbon SCFAs, is particularly conductive to colon integrity by providing the main fuel of colonocytes [13]. A previous study indicated that sodium butyrate in drinking water can ameliorate inflammatory response induced by rectal perfusion of 2, 4, 6-trinitrobenzene sulfonic acid (TNBS) [14]. Another study showed that drinking water containing butyrate reduced disease activity index (DAI) and relieved mucosal damage during dextran sulfate sodium (DSS)-induced colitis [15,16]. Nevertheless, the anti-inflammatory effects provided orally by butyrate have been less well evaluated since it is generally understood that butyrate is absorbed and utilized early, before reaching the colon [17].

Ruminococcaceae and Lachnospiraceae, two predominant families of Firmicutes in the human colon, produce the majority of butyrate [18]. Some specific intestinal microbes also improve colitis [19,20]. For example, supplementation with Akkermansia muciniphila improves colitis symptoms in mice treated with DSS [20]. Accordingly, our study aimed to determine the functions and potential mechanisms of sodium butyrate in gut microbiota dysbiosis in colitis induced by DSS in a mouse model.

## 2. Materials and Methods

### 2.1. Animals, Treatments

All the protocols used in this study were reviewed and approved by the Northeast Agricultural University Institutional Animal Care and Use Committee. All animal procedures were conducted according to the standards specified in the “Laboratory Animal Management Regulations” (revised 2016) of Heilongjiang Province, China. Forty male C57 mice (6 weeks old) were purchased from the Liao Ning Chang Sheng Biotechnology Co., Ltd. (Benxi, China) and reared in a managed animal room in which the temperature was controlled at 22–24 °C with 50–60% humidity, a 12 h light/dark cycle, and with water and a basal diet freely available. A total of 40 mice were randomly assigned to 3 experimental groups (n = 12), which received the standard diet (1. Con group); the standard diet plus DSS at a final concentration of 4% (wt/vol) (molecular weight (MW) 40 kDa: MP Biomedicals, Soho, OH, USA) in the drinking water for 5 days starting on the 29th day (2. DSS group); or oral gavage of 3% NaB and 4% DSS in the drinking water (3. NaB + DSS group). Then, these mice were euthanized via cervical dislocation. The colonic tissues were immediately fixed in formaldehyde solution for intestinal morphological detection, and the cecum contents were snap-frozen in liquid nitrogen after sampling, then frozen at −80 °C until microbiota analysis.

### 2.2. DAI and Histological Evaluation

The DAI was determined based on the scores recorded after DSS drinking, including weight loss, stool consistency, and fecal blood scores. The scoring system is shown in Table 1 [21]. A histological evaluation (H&E) of the colon and spleen was performed as described previously [22]. We evaluated histological grading according to the inflammatory criteria (Table 2) [23].

### 2.3. Microbiome Profiling and Analysis (Gut Microbiota Determination)

For microbe analysis, metagenomic DNA extraction from the cecum contents was conducted by using the PowerFecal DNA visolation kit (Mo Bio Laboratories, Inc., Carlsbad, CA, USA) according to the manufacturer’s recommendations. The quantification of DNA was performed with a NanoDrop spectrophotometer (Thermo Scientific, Waltham, MA, USA). The 16S rRNA gene marker was amplified using universal primers tailed with Illumina barcoded adapters. The PCR products and their sizes were verified using the Qubit dsDNA BR assay kit (Life Technology, Carlsbad, CA, USA) and the DNA analysis ScreenTape assay was performed on a Tape Station 2200 (Agilent Technologies, Santa Clara, CA, USA). Then, another round of PCR was performed to incorporate index sequences using the Nextera XT Index Kit (Illumina, San Diego, CA, USA). The final PCR products were cleaned and pooled. Thereafter, they were equally diluted to 4 nM and eventually denatured by using NaOH. Subsequently, the amplification products were paired end sequenced on a MiSeq platform (Illumina, Inc., San Diego, CA, USA). We calculated alpha and beta diversity to verify species diversity. The distance matrices were measured by principal coordinate analysis (PCoA) in Origin 2019b [24]. 

### 2.4. Statistical Analysis

The sequencing data were analyzed using the Quantitative Insights into Microbial Ecology (QIIME, V.1.9.1) pipeline. Taxonomic analysis based on the 16S rRNA genetic database was conducted according to the linear discriminant analysis (LDA) effect size [25]. Statistical comparisons were conducted either by one-way analysis of variance (ANOVA) followed by Tukey’s post hoc test or by the Kruskal−Wallis test followed by Dunn’s post hoc test.

## 3. Results

### 3.1. NaB Ameliorates Colitis Symptoms

Initially, to define whether NaB played a protective role against colitis in vivo, a series of clinical signs of colitis were monitored in the DSS-induced mice, including their weight, stool consistency, and rectal bleeding. Initially, the survival rate was not significantly different between the DSS-induced mice pretreated with NaB for 21 days and the non-pretreated mice, and no mice were euthanized during DSS treatment (data not shown). On day 21, an examination of the length and histology of the experimental mice’s colons was performed, which demonstrated that DSS administration significantly reduced the length of the colon. Relative to the DSS group, oral NaB effectively restored colon length (Figure 1A,B). Additionally, NaB treatment significantly reduced the DAI score after DSS challenge (Figure 1C). The increase in spleen weight was also an index of the degree of colitis severity (Figure 1D). In agreement with the alleviation of clinical symptoms, the NaB group presented less histological injury, with a reduction in the denudation and ulceration of colon epithelial cells (Figure 1E). Notably, H&E staining indicated a marked decrease in inflammatory cell infiltration in the spleen of NaB-group mice (Figure 1F). The histological scores further confirmed that oral NaB administration decreased the extent of colitis. (Figure 1G). Therefore, NaB protection from DSS-induced colitis was verified.

### 3.2. Dietary NaB Increased Gut Microbiota Diversity to Alleviate Colitis

To assess whether dietary treatment with NaB maintained gut microbiome diversity to alleviate DSS-induced colitis, we performed high-throughput sequencing to evaluate the cecum content microbiome in the mice subjected to the different treatments. For sequence analysis and quality filtering, the pyrosequencing results for the bacterial 16S rRNA genes across all 9 (n = 3) samples were first assessed. After denoising, the removal of chimeras, and the filtering of low-quality sequences, 322,940 high-quality sequences with reads longer than 350 bp were regarded as representing bacteria at a 97% similarity level. The length of the quality sequences per sample was 375.07 nucleotides after filtering.

Then, the within-community (α) diversity indices at four taxonomic levels, including the Chao 1 estimator and Ace index for community richness (Figure 2A,B) and the Shannon index and Simpson index for community diversity (Figure 2C,D), were assessed. The rarefaction curves generated by Mothur software plotting the number of OTUs tended to reach a plateau (data not shown), suggesting that the sequence depth was sufficient for capturing the majority of OTUs in the cecum samples. According to these data, NaB treatment significantly increased α diversity in the colitic mice.

To assess the β-diversity between pairs of gut microbial communities, we applied principal coordinate analysis (PCoA, which was based on weighted algorithms). The PCoA results revealed that the enterotypes of the cecum microbiota were inextricably related to the three robust clusters in the control, DSS, and NaB+DSS groups (n = 3) (Figure 2E). There were significant alterations in microbial community structure in the DSS group; however, oral NaB markedly changed β-diversity in the colitic mice (Figure 2E). ANOSIM was further performed to evaluate the significance of separation with weighted UniFrac distances among the 3 groups. 

### 3.3. Dietary NaB Ameliorated Gut Microbial Dysbiosis to Alleviate Colitis 

The alleviation of NaB in colitic mice induced by DSS challenge was accompanied by the restoration of microbial diversity and the microbial composition and interactions. In the present study, we performed LEfSe analysis to examine the differences in the gut microbiota among the three groups (Figure 3A), and linear discriminant analysis (LDA) revealed the most differentially enriched taxa in the Con, DSS, and NaB+DSS groups (Figure 3B).

At the phylum level, Bacteroidetes and Firmicutes were two most prevalent phyla in the mouse cecum samples after the Con, DSS, and NaB + DSS treatments, followed by Deferribacteres. Amounts above three phyla accounted for more than 95% bacteria in the cecum contents (Figure 4A). DSS administration increased the abundance of Bacteroidetes but decreased the abundance of Firmicutes in the treated mice compared with that of the control mice (Figure 4A). Additionally, after oral NaB, there was a decrease in the relative abundance of Bacteroidetes and Proteobacteria in the mice that received DSS, and Firmicutes were more abundant. Notably, NaB changed the Firmicutes/Bacteroidetes ratio from 0.69 to 0.89 in this study after DSS administration. In the DSS group, the expressions of Deferribacteres, Cyanobacteria, Tenericutes, and Proteobacteria were also higher; however, the expression of Actinobacteria was lower than that of the control group. Relative to the Con and DSS groups, there was a higher abundance of Cyanobacteria, Deferribacteres, and Tenericutes in the NaB-treated mice. Together, these results demonstrated that oral NaB had beneficial effects on the dysbiosis of the gut microbiota caused by DSS treatment at the phylum level.

At the class level, Bacteroidia and Clostridia were the most prevalent classes in all treatment groups, followed by Deferribacteres (Figure 4B). Colitic mice showed a higher abundance of Bacteroidia, Deferribacteres, Erysipelotrichia, Betaproteobacteria, Melainabacteria, Mollicutes, and Alphaproteobacteria and a lower abundance of Clostridia, Bacilli, and Coriobacteriia than the control mice (Figure 4B). Compared with colitic mice, NaB decreased the relative abundance of Bacteroidia, Alphaproteobacteria, Betaproteobacteria, and Erysipelotrichia and increased the proportion of Clostridia. However, the relative abundance of Deferribacteres, Melainabacteria, and Mollicutes was higher, and the relative abundance of Bacilli Coriobacteriia was lower in the NaB+DSS group than in the other experimental groups.

At the order level, the colitic mice showed a higher abundance of Bacteroidales, Deferribacterales, Burkholderiales, Erysipelotrichales, Gastranaerophilales, Mollicutes RF9, and Rhodospirillales and a lower abundance of Clostridiales, Coriobacteriales, and Lactobacillales than the control group of mice (Figure 4C). Conversely, compared with the colitic mice, feeding with NaB decreased the relative abundance of Bacteroidales, Erysipelotrichales, and Burkholderiales and increased the proportion of Clostridiales. However, compared with that in the Con and DSS groups, the relative abundance of Deferribacterales, Gastranaerophilales, and Mollicutes RF9 was increased, and the relative abundance of Coriobacteriales and Lactobacillales was decreased in the NaB + DSS group.

At the family level, Bacteroidaceae and the Bacteroidales S24-7 group were the most prevalent families, followed by Lachnospiraceae and Ruminococcaceae (Figure 4D). Colitic mice showed a higher abundance of Bacteroidaceae, the Bacteroidales S24-7 group, the Clostridiales vadinBB60 group, Deferribacteraceae, Erysipelotrichaceae, Gastranaerophilales_norank, Mollicutes RF9_norank, Porphyromonadaceae, Prevotellaceae, Rhodospirillaceae, and Ruminococcaceae and a lower abundance of Coriobacteriaceae, Defluviitaleaceae Family XIII, Lachnospiraceae, Lactobacillaceae, Peptococcaceae, and Rikenellaceae than the non-colitic control mice (Figure 4D). Infusion with NaB decreased the relative abundance of Bacteroidaceae, Ruminococcaceae, Prevotellaceae, Erysipelotrichaceae, Alcaligenaceae, and Rhodospirillaceae and increased the proportion of Lachnospiraceae. However, a higher relative abundance of the Bacteroidales S24-7 group, the Clostridiales vadinBB60 group, Deferribacteraceae Gastranaerophilales_norank, Mollicutes RF9_norank, and Porphyromonadaceae and lower relative abundance of Coriobacteriaceae and Family XIII, Lactobacillaceae, Peptococcaceae, and Rikenellaceae were observed in the NaB + DSS group than in the other experimental groups.

Then, bacterial abundance at the genus level was analyzed in detail (Figure 4E). In the colonic fecal samples, 94 genera belonging to 9 phyla were identified. The gut bacteria were dominated by the Prevotellaceae NK3B31 group, Alistipes, the Lachnospiraceae NK4A136 group, Bacteroides, Lachnoclostridium, and several unknown genera in the f__Bacteroidales S24-7 group and f__Lachnospiraceae families, together accounting for more than 50% of the total gut bacteria.

The colitic mice showed a higher abundance of Bacteroides and Ruminiclostridium 6 and a lower abundance of Lachnoclostridium and the Lachnospiraceae NK4A136 group than the control group (Figure 5). Infusion with NaB decreased the relative abundance of Bacteroides and Ruminiclostridium 6 and increased the proportions of the Lachnospiraceae NK4A136 group and Lachnoclostridium.

## 4. Discussion

The DSS model is regarded as a useful tool for the experimental model of ulcerative colitis [26]. In the model, the colon length shortened, and the DAI score, spleen weight, and histology score increased significantly [27]. In contrast, the mice supplemented with NaB showed a decreased DAI score, spleen weight, histology score, and restored colon length. No difference in the colon length was observed between Con group and NaB group, which further supported the anti-inflammatory effect of NaB on DSS-induced colitis.

Although the pathogenesis of IBD is complex, accumulating evidence showed that the development of IBD is related to the imbalance of gut microbiota composition [28]. The present study indicate that DSS treatment caused the reduction of α diversity and the increase of relative abundance of pathogenic bacteria compared with the control group, which are consistent with the results of a previous study on human patients with colitis [29,30,31]. A recovery of α diversity and changes in fecal bacteria were observed after NaB administration. In contrast, the difference in β diversity among the three groups was not significant, suggesting that NaB could not fully restore the gut microbiota composition in the colitic mouse model [32].

It was previously reported that colitic mice exhibit a greater abundance of Bacteroidetes than healthy mice, which suggests that commensal Bacteroidetes potentially play a role in promoting inflammation [33]. Consistent with the previous results, after DSS treatment, the ratio of Firmicutes/Bacteroidetes (0.69) was markedly decreased compared with that in the control group (1.06), similar to what was observed in human IBD patients [33]. In addition, it was noteworthy that the genus abundances of Lachnoclostridium, Lachnospiraceae_NK4A136_group were increased by NaB treatment to some extent compared to the DSS group. Due to their production of short-chain fatty acids which can be utilized in colonocytes for sustenance, both the Lachnoclostridium and Lachnospiraceae_NK4A136_groups were considered as anti-inflammatory factors [34]. It has been reported that some gut bacteria are associated with intestinal inflammation [35]. The Ruminiclostridium 6 abundance was positively correlated with the cytokines, DAI, and the pathological score. Thus, the increasing abundance of Ruminiclostridium 6 could be an indicator of colitis [36]. As expected, the present findings show that NaB treatment can improve the gut microbial dysbiosis by decreasing the abundance of Ruminiclostridium 6.

Briefly, dietary NaB can alleviate inflammatory response in DSS-induced colitis by maintaining gut microbiota dysbiosis, which is evidenced by the increasing abundance of beneficial bacteria, the decreasing abundance of harmful bacteria, and the minimized impact of DSS-induced inflammation [26,33].

## 5. Conclusions

Together, these data demonstrate the ameliorative effect of oral NaB against the development of colonic inflammation in DSS-treated mice. This ameliorative effect is highly associated with realizing the balance of gut flora. In conclusion, this work reveals that oral NaB targeting gut flora provides a potential approach for preventing and treating colitis. In addition, future works are also needed to fully demonstrate the mechanisms related to the effects of NaB on colitis and clinical trials should be conducted to test its safety and efficacy.

## Figures and Tables

**Figure 1 animals-10-01154-f001:**
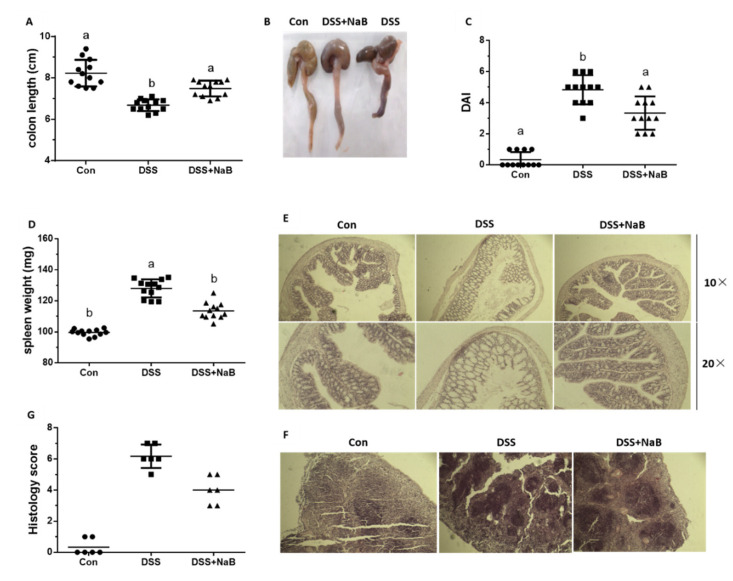
Oral NaB alleviates dextran sulfate sodium (DSS)-treated mouse colitis: (**A, B**) mouse colon length and morphology (*n* = 8 per group), (**C**) DAI (*n* = 12 per group), (**D**) spleen weight (n = 12 per group), (**E, F**) histological examination of colon tissues, and the spleen (*n* = 6 per group) with histological evaluation (H&E) staining (magnification of 10×, 20×), (**G**) histological score (*n* = 6 per group). Con, control treatment; DSS, DSS treatment under the control treatment conditions; NaB + DSS, dietary addition of NaB followed by DSS treatment. Data are shown as the mean ± standard deviation (SD), and values with different letters show significant differences (*p* < 0.05).

**Figure 2 animals-10-01154-f002:**
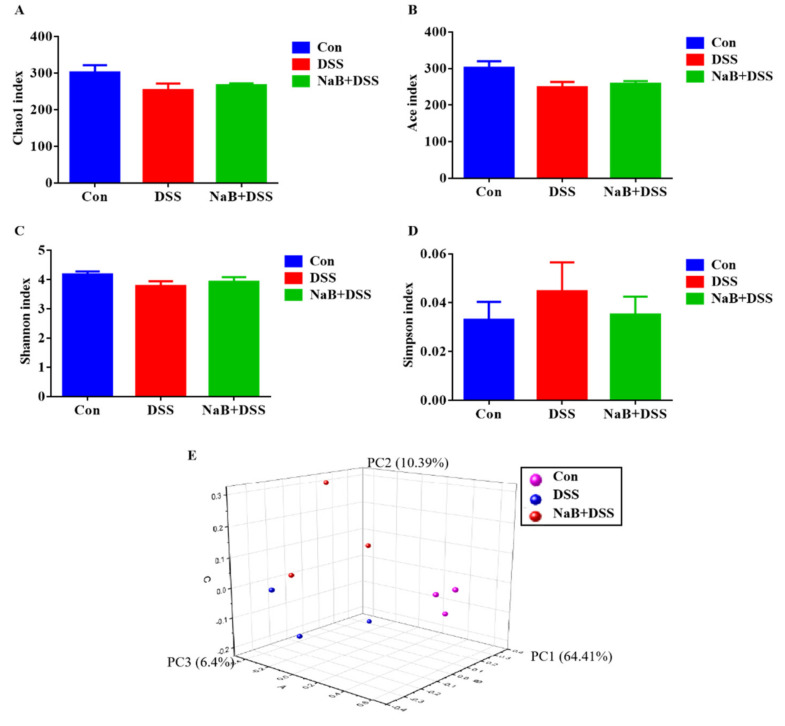
NaB altered microbial diversity in the colitic mice. The alpha diversity indices included the (**A**) Chao 1 estimator; (**B**) ACE index; (**C**) Shannon index; (**D**) Simpson index; and (**E**) PCoA of β diversity of cecal bacteria. Con, control treatment; DSS, DSS treatment under the control treatment conditions; NaB + DSS, dietary addition of NaB followed by DSS treatment. Values are shown as the mean ± SD or the median with the 95% confidence interval (*n* = 5).

**Figure 3 animals-10-01154-f003:**
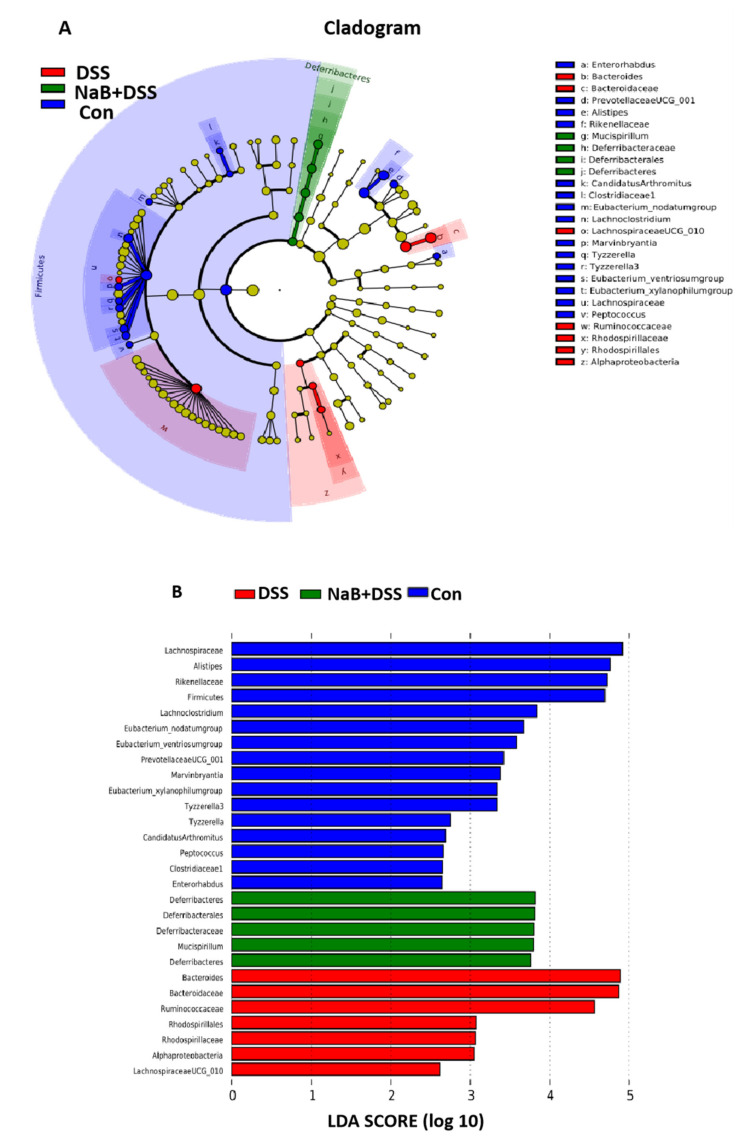
LEfSe analysis of the microbiota: (**A**) taxonomic cladogram generated from the LEfSe analysis of 16S rRNA sequences and (**B**) bacterial taxa that met the criterion of an LDA score >2 and were considered biomarker taxa.

**Figure 4 animals-10-01154-f004:**
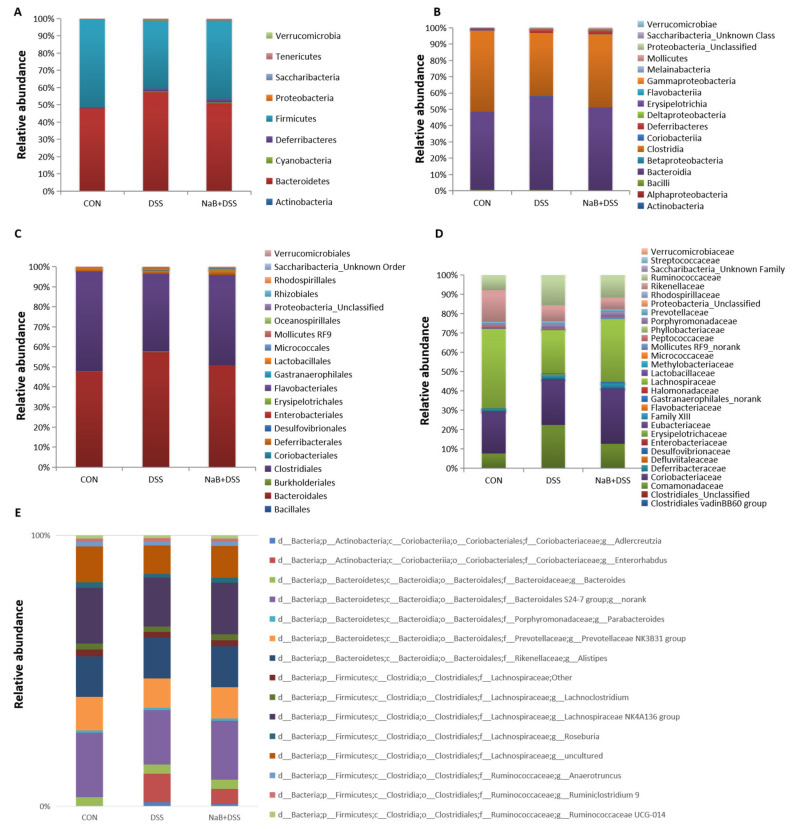
Analysis of taxa at the (**A**) phylum level; (**B**) class level; (**C**) order level; (**D**) family level; and (**E**) genus level among the different treatment groups.

**Figure 5 animals-10-01154-f005:**
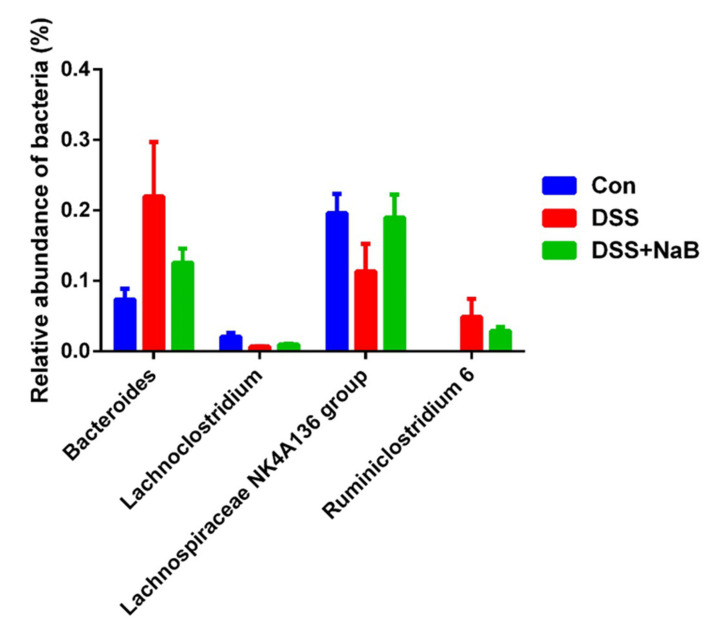
Relative abundance of selected taxa at the genus level. Data are shown as the means ± SD. Different letters represent statistically significant differences among different treatment groups (*p* < 0.05).

**Table 1 animals-10-01154-t001:** Scores of the disease activity index (DAI).

Score	Body Weight Loss (%)	Stool Consistency	Fecal Blood
0	None	Normal	No blood
1	1–5	Soft but firm	Hemoccult+
2	5–10	Loose stools	Blood
3	10–20	Diarrhea	Gross blood
4	>20		

**Table 2 animals-10-01154-t002:** Histological scoring of tissues.

Score	Inflammation Severity	Inflammation Extent	Crypt Damage
0	None	Normal	None
1	Mild	Mucosal	Basal 1/3
2	Moderate	Mucosal and submucosal	Basal 2/3
3	Severe	Transmural	Crypts lost but surface epithelium intact
4			Crypts and surface epithelium lost

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
