# Peer review of "Sodium Butyrate Alleviates Mouse Colitis by Regulating Gut Microbiota Dysbiosis"

_animals, 2020, doi:10.3390/ani10071154_

Round 1
Reviewer 1 Report
The manuscript «Sodium Butyrate Alleviates Mouse Colitis by Regulating Gut Microbiota Dysbiosis» by Xiujing Dou, Di Yan, Nan Gao and Anshan Shan is devoted to an interesting and important field at the “cross-section” of animal biochemistry, gut microbiota dysbiosis, inflammatory bowel disease ( such as colitis), mouse models for human medicine, etc. The author’s main aim was to determine an influence of sodium butyrate (by oral admission) on the relieve of DSS-induced colitis in mice (by using H&E staining technology and 16S rRNA sequence analysis). The authors found that “the severity of colitis in mice receiving oral sodium butyrate was reduced”, as well as “the composition of gut microbiota was changed”. These data proved that sodium butyrate can relieve DSS-induced colitis in mice by restoring the balance of gut microbiota dysbiosis. I do not doubt the technical quality of the work and feel that there is a sufficient impact on a broader readership to justify publication in the "Animals".
I have some comments on the manuscript. There are some errors in usage of English that I understand might be due to English not being the first language of the authors.
- There are some “gaps” between 3 rather short parts in the “Introduction” that can be filled with additional literature works. The authors must discuss more “deeply” some previous findings concerning the various ways of butyrate admission and the anti-inflammatory effects of butyrate by analyzing the additional literature works in detail. For example, the following references are missing: 1) Guangxin Chen, Xin Ran, Bai Li, Yuhang Li, Dewei He, Bingxu Huang, Shoupeng Fu, Juxiong Liu, and Wei Wang "Sodium Butyrate Inhibits Inflammation and Maintains Epithelium Barrier Integrity in a TNBS-induced Inflammatory Bowel Disease Mice Model” EBioMedicine. 2018 Apr; 30: 317–325. doi: 10.1016/j.ebiom.2018.03.030 ; 2) Jian Ji, Dingming Shu, Mingzhu Zheng, Jie Wang, Chenglong Luo, Yan Wang, Fuyou Guo, Xian Zou, Xiaohui Lv, Ying Li, Tianfei Liu & Hao Qu Microbial metabolite butyrate facilitates M2 macrophage polarization and function. Scientific Reports Published: 20 April 2016, 6:24838 . DOI: 10.1038/srep24838 .
- There is a typical overlap between results and discussion section. Please try to keep them separate as “3. Results” and “4.Discussions”.
- It is important to clarify the novelty of this study in the discussion section.
- I am sorry to say that conclusions section is poorly written and needs to be carefully expanded. Just a minor suggestion is to suggest the future direction of this research in the conclusion section.
I suggest major revision of the manuscript before recommending for publication.
Reviewer 2 Report
Inflammatory bowel disease is a very important research topic in human and veterinary medicine. This study proofs the protective effects and underlying mechanisms of a short-chain fatty acid salt, sodium butyrate, on colonic inflammation induced by dextran sulfate sodium in mice.
I can recommend the current form of the manuscript for publication.
Reviewer 3 Report
This study examined the effect of sodium butyrate as a material to ameliorate the development of IBD due to DSS, and although the manuscript was relatively well written, I felt that some revisions were needed.
General Comments.
Why was the analysis of bacteria only done with the minimum number (n=3) required for statistical processing?
In introduction, Lachnospiraceae produces butyric acid was described, so the reader will understand, but the results and discussion are notable for the notation that some bacteria have increased and decreased, and the connection to the description of improved gut health in the last paragraph seems weak.
Doesn't the Animals regulation state results and discussion separately?
Minor corrections.
Figure descriptions are notated below Figure.
The names of microorganisms should not be italicized except for genus and species.
Round 2
Reviewer 1 Report
The authors of the manuscript animals-850280 did a great work to improve all parts of this manuscript. The manuscript is devoted to an interesting and important field at the “cross-section” of animal biochemistry, gut microbiota dysbiosis, inflammatory bowel disease ( such as colitis), mouse models for human medicine, etc.
It is important to highlight the author’s improvements in the parts “1 Introduction”, “3 Results” and “4 Discussions”.
I have just one additional comment on the manuscript. It can be useful to add the words “laboratory animals” in the “Keywords” section.
I suggest to accept this manuscript in the present form.